

# Towards an in-depth characterization of Symbiodiniaceae in tropical giant clams via metabarcoding of pooled multi-gene amplicons

Xavier Pochon[1,2], Patricia Wecker[3], Michael Stat[4], Véronique Berteaux-Lecellier[5] and Gaël Lecellier[5,6]

[1] Coastal & Freshwater Group, Cawthron Institute, Nelson, New Zealand
[2] Institute of Marine Science, University of Auckland, Auckland, New Zealand
[3] Independent consultant, Montreuil sur mer, France
[4] School of Environmental and Life Sciences, The University of Newcastle, Callaghan, NSW, Australia
[5] UMR250/9220 ENTROPIE, IRD-CNRS-UR, LabEx CORAIL, Nouméa, New-Caledonia
[6] Université Paris-Saclay, UVSQ, Versailles Cedex, France

## ABSTRACT

High-throughput sequencing is revolutionizing our ability to comprehensively characterize free-living and symbiotic Symbiodiniaceae, a diverse dinoflagellate group that plays a critical role in coral reef ecosystems. Most studies however, focus on a single marker for metabarcoding Symbiodiniaceae, potentially missing important ecological traits that a combination of markers may capture. In this proof-of-concept study, we used a small set of symbiotic giant clam (*Tridacna maxima*) samples obtained from nine French Polynesian locations and tested a dual-index sequence library preparation method that pools and simultaneously sequences multiple Symbiodiniaceae gene amplicons per sample for in-depth biodiversity assessments. The rationale for this approach was to allow the metabarcoding of multiple genes without extra costs associated with additional single amplicon dual indexing and library preparations. Our results showed that the technique effectively recovered very similar proportions of sequence reads and dominant Symbiodiniaceae clades among the three pooled gene amplicons investigated per sample, and captured varying levels of phylogenetic resolution enabling a more comprehensive assessment of the diversity present. The pooled Symbiodiniaceae multi-gene metabarcoding approach described here is readily scalable, offering considerable analytical cost savings while providing sufficient phylogenetic information and sequence coverage.

# INTRODUCTION

Giant clams (Family Tridacnidae) play important roles in reef systems, acting as shelter for a number of organisms (*Cabaitan, Gomez & Aliño, 2008*; *Mercier & Hamel, 1996*), contributing to primary production through their symbiosis with dinoflagellates

Corresponding author
Xavier Pochon,
xavier.pochon@cawthron.org.nz

(*Neo et al., 2015*), and as effective filter feeders (*Klumpp & Griffiths, 1994*). Due to their large size, relative abundance and longevity, giant clams can be considered as centennial barometers of reef health (*Knop, 1996*; *Romanek & Grossman, 1989*; *Watanabe et al., 2004*). Unfortunately, as a highly prized resource throughout much of their Indo-Pacific range, the family Tridacnidae also contains some of the most endangered species due to overfishing, that is, wild stock depletion and local extinctions (*IUCN Red List, 2018*). This collapse is compounded with habitat degradation (*Bin Othman, Goh & Todd, 2010*).

Giant clams on shallow reefs allow for the establishment of a diverse in-situ reservoir of interacting fungal, bacterial, and micro-algal communities (*Baker, 2003*; *Neo et al., 2015*). Importantly, they form obligatory symbioses with, and release living cells of, Symbiodiniaceae sensu *LaJeunesse et al. (2018)*, a group of dinoflagellates that are critical for the survival of a myriad of tropical invertebrates, including corals. Despite these dynamic interactions, very little is known about the extent of symbiont diversity within giant clams and the potential exchange with other reef invertebrates engaged in similar symbiotic associations (e.g., nudibranchs and corals; *Wecker et al., 2015*). Unlike traditional molecular techniques (e.g., Polymerase chain reactions (PCR)-based fingerprinting methods and Sanger sequencing) that have been extensively used to shed light on Symbiodiniaceae diversity in reef organisms (reviewed in *Coffroth & Santos, 2005*; *Stat, Carter & Hoegh-Guldberg, 2006*), recent advances in high-throughput sequencing (HTS) technologies now enable unprecedented sequencing depth for global biodiversity assessments of symbiotic and free-living communities of Symbiodiniaceae (*Boulotte et al., 2016*; *Cunning et al., 2015*; *Edmunds et al., 2014*; *Hume et al., 2018*; *Shinzato et al., 2018*; *Thomas et al., 2014*). Nevertheless, such studies usually focus on metabarcoding analyses of single molecular markers in isolation, in particular the Internal Transcribed Spacer 2 (*ITS2*) marker (but see *Smith, Ketchum & Burt, 2017*; *Thomas et al., 2014*), potentially overlooking intrinsic phylogenetic differences known to occur between distinct Symbiodiniaceae genes (*Pochon et al., 2012*; *Pochon, Putnam & Gates, 2014*).

A variety of HTS library preparation methods exist for metabarcoding biological samples using Illumina$^{TM}$ (San Diego, CA, USA) sequencing platforms, including the use of fusion tag primers (*Elbrecht & Steinke, 2018*; *Stat et al., 2017*), the ligation of Illumina$^{TM}$ adapters using TruSeq$^{TM}$ PCR-free kits (*Rhodes, Beale & Fisher, 2014*), and the addition of Illumina$^{TM}$ adapters via dual-index sequencing (*Kozich et al., 2013*). The latter technique requires two distinct rounds of PCR analyses. The first round uses gene-specific primers modified to include Illumina$^{TM}$ adapter tails. Following purification of the PCR products, a second short round of PCR is applied using Nextera$^{TM}$ library construction kits that involve individual primer sets containing the Illumina$^{TM}$ adapter and sequencing primer sequence. This second PCR step is usually performed on individual PCR amplicon products before the pooling and sequencing of multiple samples so that demultiplexing of sequence data results in appropriate identification of input samples. For laboratories that use the services of external genomic facilities for the preparation of their dual-index libraries, an increased sample set usually correlates positively with the analytical cost due, in part, to the use of additional Nextera$^{TM}$ indexed primers. Therefore, one solution for reducing costs when performing multi-gene analyses of individual samples, is to

pool the PCR amplicon products prior to the second PCR step, followed by the sequencing and gene-specific demultiplexing per sample.

Here, we conducted a preliminary assessment of a dual-index multi-gene metabarcoding approach via the pooling and side-by-side HTS analysis of PCR amplicons from three commonly employed nuclear and chloroplastic Symbiodiniaceae markers. The ability to combine multiple gene amplicon targets per sample offers considerable analytical cost savings while providing sufficient phylogenetic information and sequence coverage. This study describes a multi-marker metabarcoding approach using giant clam *Tridacna maxima* as a model and discusses future applications for improving analyses of coral reef holobionts.

## MATERIAL AND METHODS

### Sample collection and DNA extraction

For this study, 12 DNA extracts from *T. maxima* biopsies, previously collected between February 1st 2011 and November 2nd 2013 from nine islands in the French Polynesian Archipelagos (Fig. 1; Table S1) were used (*Dubousquet et al., 2018*).

### Preparation of pooled amplicons high-throughput sequencing libraries

Three sets of Symbiodiniaceae-specific primers with Illumina™ adapter tails (Table S2) were used to amplify each sample (S141–S152; Table 1) in separate PCR. Three markers were amplified: (i) *ITS2* of the nuclear ribosomal RNA array using primers ITSD_illu and ITS2rev2_illu, (ii) the D1–D2 region of the 28S large subunit (*LSU*) nuclear ribosomal RNA gene using the newly designed primers LSU1F_illu and LSU1R_illu, and (iii) the hyper-variable region of the chloroplast 23S (*23S*) ribosomal RNA gene using primers 23SHyperUP_illu and 23SHyperDN_illu (*Manning & Gates, 2008*; *Pochon et al., 2010*). The new forward and reverse *LSU* primers were designed within the conserved areas flanking the D1–D2 region of a previously published LSU sequence alignment (*Pochon et al., 2012*; Fig. S1), containing 93 sequences of Symbiodiniaceae (with representatives from all nine existing clades), as well as eight sequences from three dinoflagellate species represented by *Gymnodinium simplex*, *Pelagodinium beii*, and *Polarella glacialis*. Primers were designed to be "dinoflagellate-specific" using MacVector v11.0.2 (MacVector Inc., Cary, NC, USA), avoiding cladal bias and minimizing self/duplex hybridization and internal secondary structure problems (Fig. S1).

Polymerase chain reactions were performed for each sample and for each gene separately in 50 µL volumes, with the reaction mixture containing 45 µL of Platinum PCR SuperMix High Fidelity (Life Technologies, Carlsbad, CA, USA), 10 µM of each primer, and 10–20 ng of template DNA. In order to maximize specificity to Symbiodiniaceae, a touchdown PCR protocol was used for each reaction as follows: (i) 95 °C for 10 min; (ii) 25 cycles of 94 °C for 30 s, 65 °C for 30 s (decreasing the annealing temperature 0.5 °C for every cycle after cycle (1), and 72 °C for 1 min; (iii) 14 cycles of 94 °C for 30 s, 52 °C for 30 s and 72 °C for 1 min; and (iv) a final extension of 72 °C for 10 min. Amplicons of the correct size (estimated visually via gel electrophoresis) were purified using Agencourt AMPure XP PCR Purification beads following the manufacturers' instructions.

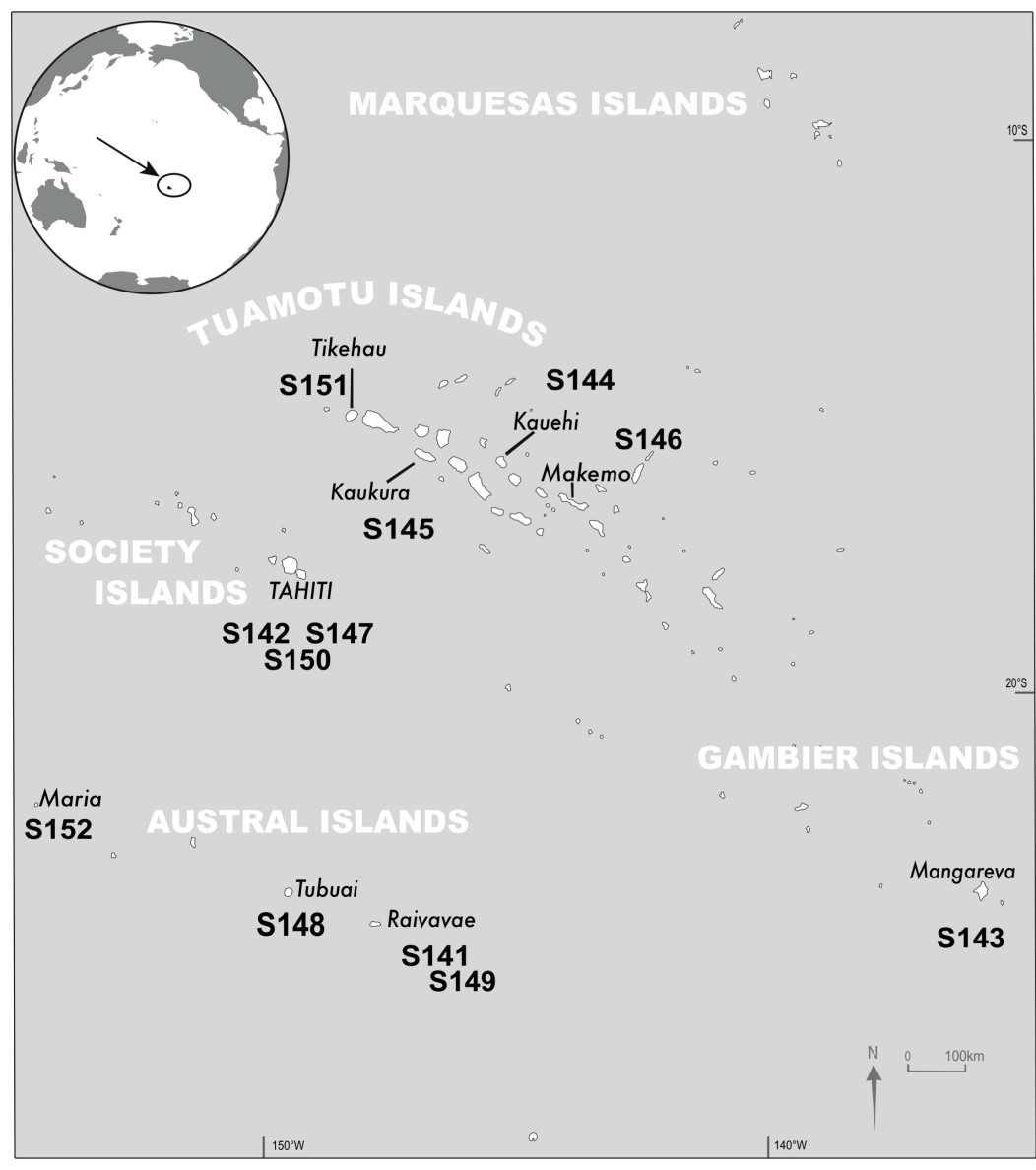

**Figure 1 Sampling sites.** Location and sample identification for the 12 *Tridacna maxima* samples investigated in this study (credit to R. Canavesio).

In order to sequence the three gene amplicons per sample collectively using HTS, individual purified products for each marker originating from the same giant clam were pooled together to enable the attachment of the same Nextera™ index (i.e., 12 samples). This was achieved by quantifying the amplicons using a Qubit Fluorometer 2.0 (Life Technologies, Carlsbad, CA, USA), diluting to one ng/μL using Milli-Q water and mixing five μL of each gene amplicon from the same giant clam together. To assess the levels of cross-contamination between samples potentially arising during the library indexing step, nine unmixed amplicon products (i.e., *ITS2*, *LSU*, and *23S* amplicons from three haphazardly selected giant clams; samples S141–S143; Table 1), each with their own unique index to be added, were also prepared.

**Table 1** DNA sequence counts following demultiplexing of "Pooled" and "Controls" samples.

| Sample ID | Source reads | Filtered reads | 23S reads | ITS2 reads | LSU reads |
|---|---|---|---|---|---|
| Pooled | | | | | |
| S141 | 75,731 | 53,654 | 22,072 | 17,813 | 13,435 |
| S142 | 89,975 | 65,312 | 26,504 | 24,395 | 14,040 |
| S143 | 78,009 | 48,881 | 21,061 | 10,256 | 17,321 |
| S144 | 1,72,319 | 1,26,860 | 48,941 | 39,131 | 38,128 |
| S145 | 1,47,293 | 1,04,743 | 31,048 | 34,457 | 38,662 |
| S146 | 72,548 | 51,886 | 23,268 | 16,817 | 11,537 |
| S147 | 1,18,815 | 79,339 | 29,870 | 32,449 | 16,332 |
| S148 | 50,176 | 34,810 | 12,577 | 11,695 | 10,264 |
| S149 | 4,728 | 3,381 | 2,400 | 366 | 599 |
| S150 | 88,926 | 59,387 | 20,788 | 22,068 | 16,216 |
| S151 | 53,016 | 38,314 | 15,964 | 12,882 | 9,298 |
| S152 | 60,107 | 42,239 | 17,075 | 13,108 | 11,707 |
| Controls ITS2 only | | | | | |
| S141 | 85,824 | 52,588 | 8 | 52,335 | 1 |
| S142 | 81,924 | 52,270 | 10 | 51,988 | 6 |
| S143* | 130 | 13 | 5 | 6 | 2 |
| LSU only | | | | | |
| S141 | 56,565 | 31,134 | 8 | 7 | 30,758 |
| S142 | 92,110 | 62,629 | 23 | 0 | 62,129 |
| S143 | 1,14,431 | 69,823 | 9 | 0 | 69,318 |
| 23S only | | | | | |
| S141 | 77,522 | 66,763 | 66,399 | 3 | 3 |
| S142 | 42,004 | 36,422 | 36,263 | 3 | 9 |
| S143 | 27,894 | 24,239 | 24,149 | 1 | 3 |
| Total reads | 1,590,047 | 1,104,687 | 398,442 | 339,780 | 359,768 |

Notes:
Number of DNA sequences recovered from each sample (S141–S152), before and after quality filtration, and after demultiplexing into each gene. Samples S141–S143 were used as control samples, each targeting only one of three PCR amplicons. Columns highlighted in gray show a low background contamination.
* One control sample (S143 ITS2) failed at sequencing, resulting in only 130 raw reads.

The resulting 21 samples were placed on a 96-well plate along with other samples published elsewhere (*Zaiko et al., 2016*), and sent to New Zealand Genomics Ltd. (University of Auckland, Auckland, New Zealand) for HTS library preparation which involved a second round of PCR to attach the Nextera[TM] indexes on to the amplicons for MiSeq Illumina[TM] sequencing. PCR products were combined in equimolar concentrations and the final library paired-end sequenced on an Illumina[TM] MiSeq using a 500 cycle (2 × 250) MiSeq® v2 Reagent Kit and standard flow cell.

## Bioinformatics

Illumina[TM] sequence datasets were prepared using the read preparation and dereplication pipeline of USEARCH (*Edgar, 2010*). Firstly, paired reads were merged

(fastq_mergepairs command) and filtered (fastq_filter command) with an expected number of error of 0.25. More than 90% of the base pairs had a Q score >40. Next, samples were demultiplexed in three groups, primers were trimmed and a global trimming was operated according to the recommendations for *ITS* amplicon reads (*Edgar, 2013*). The sequence data were dereplicated and unique singletons found across the complete dataset were discarded.

For phylogenetic assignments of Symbiodiniaceae, three distinct annotated reference databases (*ITS2*, *LSU*, and *23S*) were generated in fasta format, including sequence representatives from each of nine Symbiodiniaceae clades (A–I), with (i) 409 representative *ITS2* phylotypes from GeoSymbio (*Franklin et al., 2012*), (ii) 37 representative *LSU* sequences from *Pochon et al. (2012)*, and (iii) 104 sequences of *23S* from *Takabayashi et al. (2011)*. The three reference sequence databases used in the present study are provided in the File S1. Symbiodiniaceae assignments were performed using the software "Kallisto" (*Bray et al., 2016*) which provides speed and accuracy for optimal analysis of large-scale datasets (e.g., large RNA-Seq data) without the need for time-consuming alignment steps.

Because the main goal of the present pilot study was to investigate the sequencing depth and potential inter-marker biases of the multi-marker metabarcoding approach using giant clam samples as a proof-of-concept, as opposed to describing potentially novel Symbiodiniaceae diversity in these samples, we modified the Kallisto pipeline to only retain HTS reads yielding exact matches (i.e., without ambiguity amongst k-mers) to individual referenced genotypes in each gene. Individual sequences generated via HTS were then blasted against all pseudo-alignments and exact matches against the entire population of k-mers were recorded. To reduce mis-assignments, all merged reads with ambiguities between k-mers of different reference genotypes were determined as chimeric and removed from the dataset. These sequences that did not result in exact matches could correspond to non-Symbiodiniaceae sequences or to sequences not comprised in our custom databases. Therefore, a second comparison using BLASTn (threshold: *e*-value $<10^{-30}$) against the National Center for Biotechnology Information (NCBI) nucleotide databases was performed and the accession numbers yielding exact matches were retained for downstream analyses. The number of unique sequences matching genotypes in the reference databases and GenBank was recorded (Table S3). Raw sequence data were submitted to the BioProject Archive under accession PRJNA471926 (SRR7181922–SRR7181942).

## Sequence diversity analyses

Unique sequence genotypes found at or above a 0.05% threshold from the total sequence abundance per sample were scored (Table S3) and the specific genotypes of reference (i.e., from in-house reference databases and GenBank) were retained for sequence diversity and phylogenetic analyses. Global sequence diversity from each of the three datasets (*23S*, *ITS2*, and *LSU*) were visualized using the plug-in DataBurst implemented in Excel (Microsoft Office version 2013 or later).

One sequence alignment was generated for each of the three investigated gene datasets using the sequence alignment software BioEdit v7.2.5 (*Hall, 1999*). Owing to the difficulty

in aligning sequences from *Symbiodinium* (clade A) and *Cladocopium* (clade C) genera when using the *23S* and *ITS2* genes, and between Symbiodiniaceae and non-Symbiodiniaceae (i.e., clams, fungi, and plants) sequences, phylogenetic reconstructions only aimed at depicting pair-wise relationships between retained sequence genotypes. Therefore, unrooted phylogenetic inferences were generated using the neighbor-joining method implemented in the program MEGA v. 7.0 (*Kumar, Stecher & Tamura, 2016*), with the *p*-distance model and gaps treated as pairwise deletions. Internal nodes support was tested using the bootstrap method (*Felsenstein, 1985*) and 500 replicates.

## RESULTS

A total of 1,590,047 sequences were obtained from the 21 samples (75,716 +/− 41,576 sequences per sample), which included 12 amplicon samples (S141–S152) each containing three pooled gene products (*23S*, *ITS2*, and *LSU*) and nine amplicon samples from three selected giant clam isolates (S141, S142, and S143) which only contained a single gene amplicon as internal controls (Table 1; Table S3). One sample (internal control S143 for *ITS2*) failed the sequencing step with only 130 raw reads produced. After read cleaning, the total number of high-quality sequences was 1,104,687 (52,604 +/− 29,250 sequences per sample). The proportion of total reads (Table 1) between the three investigated genes was well-balanced with 398,442 reads (*23S*), 339,780 reads (*ITS2*), and 359,768 reads (*LSU*). In contrast, unique reads varied between 23,779 sequences for the *23S* gene and 71,776 sequences for the *LSU* gene (Table S3). The inclusion of nine positive controls, representing three amplicon products per gene sequenced in isolation, revealed the presence of low levels of sequence cross-contamination between samples (mean of 4.5 sequences ± 4.6 SD) (Table 1). This low-level of background contamination (1–23 sequences per sample) represented <0.003% of the total reads per sample (Table S3). Therefore, as a conservative measure, we chose to remove sequences that represented <0.05% of the total sequence abundance per sample.

Our bioinformatics pipeline identified 43 Symbiodiniaceae *23S* chloroplast genotypes, including 16 that matched the *23S* reference database and another 27 that matched sequences in GenBank. After exclusion of genotypes represented by less than 0.05% of the sequence abundance in each sample (Table S3), the number of unique *23S* Symbiodiniaceae sequences retained for phylogenetic analysis was 11, including six sequences matching the *23S* in-house reference (Fig. S2; Table S3). Similarly, blasting *ITS2* and *LSU* datasets against both types of databases led to the identification of 117 and 93 unique sequences when using the original datasets, and to 46 and 51 unique sequences following the 0.05% filtering threshold, respectively.

Diversity diagrams were generated to visualize the sequence abundance of Symbiodiniaceae generic and sub-generic sequences recovered from the 12 giant clam samples and among the three investigated genes (Fig. 2). The pooled multi-gene approach yielded similar proportions of dominant genera, but with some notable differences. The genus *Symbiodinium* (previously Clade A) dominated in all three markers, particularly in *23S* (91.8%; dominant sub-generic sequence chvA2), with lower but similar

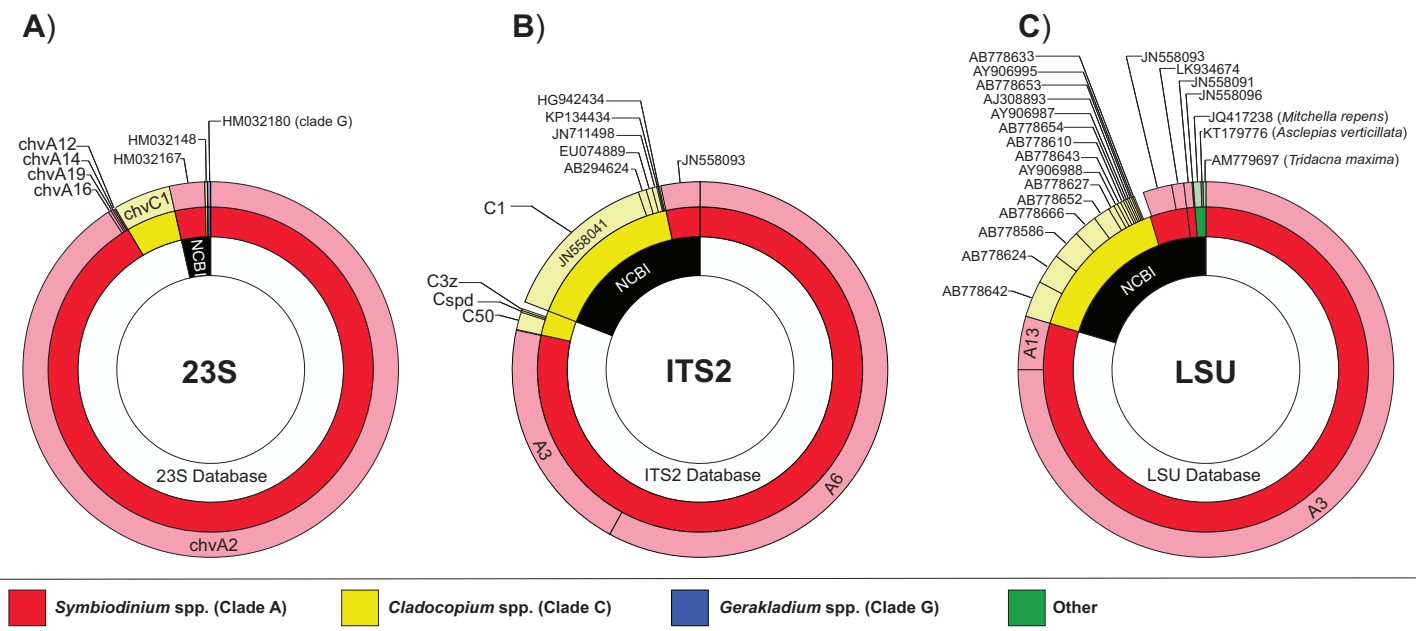

**Figure 2** **Proportion and diversity of Symbiodiniaceae genotypes across markers.** Global Symbiodiniaceae diversity charts obtained from each of the three datasets: (A) *23S*, (B) *ITS2*, and (C) *LSU*. The proportion of sequences matching one of the three in-house reference databases or NCBI (inner circles) and their corresponding phylogenetic affiliation at genus (i.e., clade; middle circles) and sub-generic (i.e., subclade; outer circles) levels. Sequence reads representing <0.1% of total read abundance are not included.

proportions between *ITS2* (81.7%; dominant sub-generic sequences A3/A6) and *LSU* (83.9%; dominant sub-generic sequences A3/A13). The genus *Cladocopium* (previously Clade C) represented 7.9% (dominant sub-generic sequence chvC1), 18.2% (dominant sequence C1), and 15.0% (dominant sequence C1) of reads for the *23S*, *ITS2*, and *LSU* markers, respectively. *Gerakladium* (previously clade G) was only detected using the chloroplast *23S* gene (0.2% of reads), whereas the nuclear *LSU* gene displayed reduced specificity for Symbiodiniaceae as indicated by ~1% of sequence reads matching other organisms such as streptophytes (*Mitchella repens* and *Asclepias verticillata*), and the host giant clam *T. maxima*. Overall, the proportion of dominant Symbiodiniaceae generic and sub-generic sequences recovered between the pooled samples and the positive (single gene) controls were very similar (Table 2).

# DISCUSSION

## Multi-gene metabarcoding: more for less

The concept of pooled multi-gene amplicons for dual-indexed metabarcoding, that is, the tagging and pooling of distinct gene amplicons before Illumina™ adapter indexing and simultaneous sequencing of samples, has been used in other research fields (*Elbrecht & Steinke, 2018*; *Keeley, Wood & Pochon, 2018*; *Marcelino & Verbruggen, 2016*; *Von Ammon et al., 2018*; *Zhang et al., 2018*), but has never been applied to Symbiodiniaceae dinoflagellates. In this proof-of-concept study, we show that the technique effectively recovered similar proportions of sequence reads and Symbiodiniaceae genera among

Table 2 Percentage comparison of each Symbiodiniaceae sub-generic genotype per gene per treatment.

| 23S (%) | chvA12 | chvA14 | chvA16 | chvA19 | chvA2 | chvC1 | HM032148 | HM032167 |
|---|---|---|---|---|---|---|---|---|
| S141 (Pooled) | 0.09 | 0.137 | 0.209 | 0.166 | 80.911 | 14.897 | 0.583 | 3.006 |
| S141 (Control) | 0.136 | 0.146 | 0.204 | 0.182 | 81.4 | 14.208 | 0.573 | 3.151 |
| S142 (Pooled) | 0.16 | 0.156 | 0.237 | 0.09 | 96.504 | 0 | 0 | 2.853 |
| S142 (Control) | 0.154 | 0.125 | 0.205 | 0.137 | 96.59 | 0 | 0 | 2.79 |
| S143 (Pooled) | 0.138 | 0.113 | 0.237 | 0.182 | 95.847 | 0 | 0 | 3.482 |
| S143 (Control) | 0.133 | 0.146 | 0.236 | 0.112 | 95.869 | 0 | 0 | 3.504 |

| ITS2 (%) | ITS_A3 | ITS_A5 | ITS_A6 | ITS_C57 | AB294624 | EU786115 | JN558041 | JN558093 |
|---|---|---|---|---|---|---|---|---|
| S141 (Pooled) | 45.912 | 0.064 | 49.321 | 0.761 | 2.497 | 0.099 | 0 | 1.347 |
| S141 (Control) | 47.037 | 0.099 | 51.951 | 0.913 | 0 | 0 | 0 | 0 |
| S142 (Pooled) | 4.281 | 0.081 | 81.403 | 0 | 0 | 0 | 0.043 | 14.191 |
| S142 (Control) | 5.177 | 0.122 | 94.701 | 0 | 0 | 0 | 0 | 0 |
| S143 (Pooled) | 96.595 | 0 | 0.146 | 0 | 0 | 0 | 0.283 | 2.975 |
| S143 (Control)* | 60 | 0 | 40 | 0 | 0 | 0 | 0 | 0 |

| LSU (%) | LSU_A13 | LSU_A3 | AB778585 | AB778586 | AM779697 | AY074967 | EU650387 | JN558091 |
|---|---|---|---|---|---|---|---|---|
| S141 (Pooled) | 2.018 | 70.821 | 0 | 10.878 | 0.737 | 0.015 | 0.357 | 1.616 |
| S141 (Control) | 2.43 | 70.47 | 0.016 | 11.051 | 0.511 | 0.013 | 0.296 | 1.711 |
| S142 (Pooled) | 18.618 | 75.42 | 0.028 | 0 | 0.036 | 0.007 | 0 | 3.134 |
| S142 (Control) | 19.001 | 74.863 | 0.031 | 0 | 0.018 | 0.018 | 0 | 3.315 |
| S143 (Pooled) | 22.83 | 72.4 | 0.023 | 0 | 0 | 0 | 0 | 3.615 |
| S143 (Control) | 23.866 | 71.319 | 0.023 | 0 | 0.001 | 0.006 | 0 | 3.71 |

| LSU (%)—continued | JN558092 | JN558093 | JN558096 | JN665089 | KC510080 | KT179776 | LK934674 |
|---|---|---|---|---|---|---|---|
| S141 (Pooled) | 0 | 0.447 | 0.007 | 0.06 | 9.657 | 2.174 | 1.214 |
| S141 (Control) | 0.01 | 0.485 | 0.023 | 0.039 | 9.539 | 2.202 | 1.204 |
| S142 (Pooled) | 0.014 | 0.264 | 0 | 0 | 0 | 0 | 2.479 |
| S142 (Control) | 0.019 | 0.261 | 0.002 | 0 | 0 | 0 | 2.474 |
| S143 (Pooled) | 0.006 | 0.173 | 0.006 | 0 | 0 | 0 | 0.947 |
| S143 (Control) | 0.009 | 0.234 | 0.001 | 0 | 0 | 0 | 0.831 |

Notes:
Percentage comparison of each Symbiodiniaceae sub-generic genotype recovered using the three amplicon markers in "Pooled" versus single "Control" markers (see Table 1). The proportion of each sub-generic type between "Pooled" and "Controls" is almost identical for the *23S* marker, but shows some minor differences for the *ITS2* and *LSU* markers (gray shades).
* One control sample (S143 ITS2 only) failed the sequencing resulting in only 130 raw reads (see Table 1).

the three pooled genes investigated per sample, providing more confidence that single gene primer biases did not occur during Nextera[TM] indexing. Another advantage is the ability to simultaneously visualize varying levels of phylogenetic resolution, enabling a more comprehensive assessment of the diversity present. For example, while the traditional "species-level" *ITS2* marker (*LaJeunesse, 2001*) enabled characterization of 46 Symbiodiniaceae sub-generic sequences, the *LSU* marker, interestingly, offered both a similarly high resolution for Symbiodiniaceae (46 sub-generic sequences) and a reduced specificity by also enabling identification of other host-associated organisms such as streptophytes, as well as the host *Tridacna*. Nevertheless, the results regarding the

streptophytes need to be interpreted with caution. Indeed, the herbaceous woody shrub *M. repens* and the milkweed *A. verticillata* are both land plants restricted to the eastern coasts of North and South America and are, therefore, highly unlikely to represent true detections from our giant clam samples. It is possible that they are either the result of PCR contamination, inaccurate annotation in GenBank and/or correspond to the next sequence hit, the plant *Coffea arabica* which has a wide distribution including the Pacific region. The hyper-variable region of the chloroplast *23S* marker used here is more conserved than the *ITS2* and *LSU* regions, but has been successfully used for specifically targeting low abundance free-living Symbiodiniaceae cells from environmental samples (*Decelle et al., 2018*; *Manning & Gates, 2008*; *Pochon et al., 2010*; *Takabayashi et al., 2011*). The unique detection of *Gerakladium* (clade G) using the 23S marker highlights the added value of the multi-gene approach for broader Symbiodiniaceae screening efficiency. This marker also showed remarkable consistency in the proportion of recovered sub-generic types between "Pooled" and "Control" samples, and contrasted with the *ITS2* and *LSU* markers (Table 2). For example, four *ITS2* sequences were detected in the "Pooled" but not in the "Control" samples, and there were five instances where *LSU* sequences were detected in the "Control" but not in the "Pooled" samples. Similarly, another difficult-to-explain contrast was observed for samples S147 and S152 (Fig. S3) where the proportion of recovered Symbiodiniaceae genera differed markedly between *23S* and *ITS2/LSU* Markers. Although the above minor differences are likely attributable to PCR or sequencing biases, further research applying similar multi-gene approaches would improve our understanding of the intrinsic characteristics of these commonly employed Symbiodiniaceae markers and help guide the interpretation of such datasets.

Analytical cost is an important consideration for any research group aiming to monitor coral reef ecosystems, and the budget needed to include HTS for biodiversity assessments is highly variable. The cost depends on the number of gene regions investigated, method of library preparation, sequencing depth, and whether pooling amplicons is employed as shown here. A comparative cost estimate between the pooling of three PCR amplicons for the 12 investigated samples versus the complete processing of thirty six individual PCR amplicons showed that the pooling method enabled an approximately 5.4 times cost saving on reagents (tubes, tips, purification/quantification, and Nextera[TM] indexing). In this context, our approach is readily scalable and has the potential to offer substantial savings in terms of both time and cost, for example, by enabling coral reefs researchers to generate multi-gene Symbiodiniaceae data in a 96-well format for the price of a single dual-indexed Illumina[TM] MiSeq run. Nevertheless, the caveat is that upscaling this pooling method beyond a certain threshold will inevitably lead to a decrease in sequencing depth per sample per gene. Exceeding this threshold may be problematic for researchers wanting to gather a complete overview of fine-scale diversity, or study potential low-frequency intragenomic variants. Further research is needed to better understand this tradeoff and to set appropriate thresholds. Another important consideration is to make sure that the distinct pooled amplicons are of similar base-pair length, otherwise shorter gene amplicons may generate more sequence reads than the longer co-occurring amplicons (*Engelbrektson et al., 2010*). Additional studies

are also required to investigate whether multiplexing that is, the mixing of multiple primer sets in the original PCR to produce multi-gene amplicons (*De Barba et al., 2014*; *Fiore-Donno et al., 2018*) would result in similar proportions of Symbiodiniaceae genotypes between markers such as shown in the present study. Such an approach, if validated, would allow very significant additional cost savings.

## Paving the way for comprehensive biodiversity assessment of giant clams

Giant clams on shallow reefs allow for the establishment of a diverse in-situ reservoir of interacting fungal, bacterial, and micro-algal communities (*Baker, 2003*; *Neo et al., 2015*). For example, they commonly harbor Symbiodiniaceae from at least three distinct genera (*Symbiodinium* (clade A), *Cladocopium* (clade C), and/or *Durusdinium* (clade D)) simultaneously or in isolation within one host, with *Symbiodinium* being the dominant symbiont genus in most clams (*Baillie et al., 2000*; *DeBoer et al., 2012*; *Ikeda et al., 2017*; *Ikeda et al., 2016*; *Pappas et al., 2017*; *Trench, Wethey & Porter, 1981*). Similar to coral symbiosis, it is assumed that the genotypic composition of Symbiodiniaceae in giant clams is influenced by environmental or physical parameters (e.g., temperature, irradiance), or by life stages and taxonomic affiliation (*Ikeda et al., 2017*; *Pappas et al., 2017*). Giant clam larvae (veliger) acquire free-living Symbiodiniaceae cells "horizontally" from their surrounding environment (*Fitt & Trench, 1981*). When mature, giant clams (e.g., *T. derasa*) expel high numbers of intact symbionts in their faeces at rates of $4.9 \times 10^5$ cells d$^{-1}$ (*Buck, Rosenthal & Saint-Paul, 2002*; *Maruyama & Heslinga, 1997*). Despite the dynamic interaction of symbionts between Tridacnidae and the environment, very little is known about the extent of symbiont diversity within giant clams and the potential exchange with other reef invertebrates engaged in similar symbiotic associations.

In this preliminary study, we found that genera *Symbiodinium* (clade A) and *Cladocopium* (clade C) dominated in adult giant clams in French Polynesia (Fig. S3). *Symbiodinium* was the major genus in our samples and in particular the closely related sub-generic *ITS2* genotypes A3 and A6, previously described as *Symbiodinium tridacniadorum*, and therefore associated with *Tridacna* clams (*Lee et al., 2015*). A3 is the most dominant genotype in *T. maxima* around the world and both A3/A6 are more likely to be sampled in giant clams from shallow reefs (*Weber, 2009*).

Furthermore, for *Cladocopium* we found that the generalist *ITS2* genotype C1 (*LaJeunesse et al., 2003*) co-dominated in our samples, which is consistent with a previous study showing C1 as a common genotype in *T. maxima* from around the world (*Weber, 2009*). We also found a smaller percentage of C3z, Cspd, and C50 *ITS2* genotypes, which to our knowledge have not yet been found in *T. maxima*, and are usually restricted to corals (*LaJeunesse et al., 2004*; *LaJeunesse et al., 2010*; *Macdonald et al., 2008*; *Shinzato et al., 2018*). Finally, we did not detect any symbiont from the genus *Durusdinium* (Clade D) despite in-depth sequencing. However, *Durusdinium* has never been detected in *T. maxima* from French Polynesia compared to other regions such as the Indian Ocean (*DeBoer et al., 2012*; *Weber, 2009*). As we only worked with adult clams from shallow water, it would be interesting to confirm the hypotheses of *Ikeda et al. (2017)*

and *Weber (2009)* who argued that *Durusdinium* symbionts might be restricted to "young" *T. squamosa* clams (less than 11 cm) or that giant clams harbored this dinoflagellate genus only when sampled from deeper reefs, respectively. Nevertheless, the small dataset used in the present study precludes us from making any relevant assumptions about potentially novel symbiont diversity in giant clams. In particular, the use of the Kallisto bioinformatics pipeline which restricted the analysis to 100% sequence similarity hits is likely not suitable for the many studies where a high degree of sequence novelty is found. Additionally, a weakness of the Kallisto method is that the analysis of k-mers that are poorly divergent and/or not well represented in the reference database may impact the final sequence annotation, in particular at the sub-clade level. For example, the *ITS2* genotype C1 was only detected following NCBI blast, even though this sequence was present in our in-house database. This is not ideal, and one could argue that sequences should have been blasted exclusively against GenBank. Nevertheless, the chosen bioinformatics pipeline did not affect the general findings of the present study and was appropriate for this purpose. It is our hope, however, that our multi-gene approach will be investigated further using a more comprehensive giant clam dataset along with the development of an alternative bioinformatics method guiding users on the assignment of genus to species-level taxon ID to novel multi-gene sequences for deposition to GenBank.

## CONCLUSIONS

This pilot project explored the use of pooled amplicon metabarcoding for rapid, cost-effective and in-depth characterization of Symbiodiniaceae dinoflagellates using the giant clam *T. maxima* as a model. Our results showed that the technique effectively recovered similar proportions of sequence reads and Symbiodiniaceae diversity among the three gene amplicons investigated per sample enabling a more comprehensive assessment of the diversity present, while also offering appreciable analytical cost savings. We also found that *Symbiodinium* (clades A) and *Cladocopium* (clade C) were the dominant genera in adult giant clams in French Polynesia, with similar sub-generic genotypes (*ITS2* A3, A6, and C1) previously described as commonly associated with giant clams from around the world. Our approach paves the way for more comprehensive surveys of this important yet endangered group of reef invertebrates and its potential role as an important Symbiodiniaceae reservoir for declining coral reefs. More work is required to test the applicability of this method to other symbiotic organisms as well as to environmental samples. Future investigations may also expand on this method to clarify species-level differentiation among Symbiodiniaceae taxa using other markers (e.g., nuclear Actin, chloroplast *psbA*), or simultaneously characterize all organisms (viruses, bacteria, fungi, and other eukaryotes) associated with a more diverse host range. Such holistic diversity assessments will improve our knowledge on the ecology and evolution of tropical holobionts and better predict the adaptation of coral reefs in a rapidly changing environment.

## ACKNOWLEDGEMENTS

We thank Jonathan Drew for his support in the laboratory, and Charley Waters, Sam Murray and Chris Cornelisen for valuable discussions during manuscript preparation.

### Funding

This work was supported by Cawthron Institute Internal Investment Fund (IIF) #BST16931 and by the French National Research Center (CNRS). The funders had no role in study design, data collection and analysis, decision to publish, or preparation of the manuscript.

### Grant Disclosures

The following grant information was disclosed by the authors:
Cawthron Institute Internal Investment Fund (IIF): #BST16931.
French National Research Center (CNRS).

### Competing Interests

Xavier Pochon is an Academic Editor for PeerJ.

### Author Contributions

- Xavier Pochon conceived and designed the experiments, performed the experiments, analyzed the data, contributed reagents/materials/analysis tools, prepared figures and/or tables, authored or reviewed drafts of the paper, approved the final draft.
- Patricia Wecker contributed reagents/materials/analysis tools, prepared figures and/or tables, authored or reviewed drafts of the paper, approved the final draft.
- Michael Stat authored or reviewed drafts of the paper, approved the final draft.
- Véronique Berteaux-Lecellier conceived and designed the experiments, prepared figures and/or tables, authored or reviewed drafts of the paper, approved the final draft.
- Gaël Lecellier conceived and designed the experiments, performed the experiments, analyzed the data, contributed reagents/materials/analysis tools, prepared figures and/or tables, authored or reviewed drafts of the paper, approved the final draft.

### Data Availability

   Raw sequence data are available at the BioProject Archive under accession PRJNA471926 (SRR7181922–SRR7181942).

### Supplemental Information

Supplemental information for this article can be found online at http://dx.doi.org/10.7717/peerj.6898#supplemental-information.

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
