# Peer review of "Towards an in-depth characterization of Symbiodiniaceae in tropical giant clams via metabarcoding of pooled multi-gene amplicons"

_PeerJ, doi:10.7717/peerj.6898_

## Round 0.1 · original submission · Minor Revisions

Three expert reviewers have evaluated your manuscript and their comments can be seen in the downloadable PDFs. All reviewers have positive comments about the utility of this methods paper. However, there are a number of issues that need to be attended, therefore I am suggesting that you revise the manuscript, making sure to attend all of the concerns that have been been raised or the suggestions that have been made to improve the manuscript.

Reviewer 1 ·

Basic reporting

This article is very well written and the methods are explained in excellent detail. The structure is clearly defined and the key findings and developments are easily accessible. Raw data is publicly available.

Experimental design

This is predominantly a methods paper, that presents a technique for highly multiplexed NGS diversity analysis for Symbiodiniaceae targeting multiple genetic markers. I believe this is a good step in the right direction, since previous studies have tended to target only a single marker due to the prohibitive costs involved. The authors demonstrate that three markers with varying taxonomic resolution can be targeted for a similar cost by pooling samples before attaching Nextera tags. This will encourage labs with limited resources to expand their methods to include to multi-gene analysis, which will ultimately help us to better-understand the diversity and evolutionary history of this group.

Validity of the findings

Experimental controls were appropriate and the laboratory analysis appears to be thorough. The new LSU primers developed in this study will be of broad utility, since the amplicon shows good specificity and taxonomic resolution. This study took the conservative approach of only assigning taxonomy in cases where a 100% similarity match was found with a previously deposited sequence. Therefore the assignments can be considered robust.

Additional comments

This is a good contribution to the field, and I look forward to adopting these methods into my own research. Thank you for the opportunity to review.

·

Basic reporting

This manuscript reports a nice little case study exploring a very promising way to explore biodiversity in Symbiodiniaceae. It is clear, and a t least from a non native English speaker point of view the English is fine enough. The literature references and structure are good.
I did not check the Raw data, I only check that the authors mention the pathway to access the raw data in the manuscript and I trust the authors did not invent this information. I believe there is no need of specialist to check access to raw data, therefore the editor can do it well enough, and if PeerJ expects reviewers to check in depth raw data, they should consider seriously paying the reviewers for the time it will take (and not give 10 days to do it).

The paper is self-contained and completely relevant to its purpose.

For more details see the general comments to the authors.

Experimental design

This research is totally suitable for PeerJ (I would even think it can go to a better journal), its question is well defined and certainly will provide useful critical information for researchers not only studying giant clams but for all the coral reef research community and potentially beyond this field too (Symbiosis, eDNA fields etc...).
The authors clearly followed high standards in Symbiodiniaceae research, although some little glitches (domains outside of Symbiodiniaceae) appear here and there but nothing that cannot be modified/discussed easily.
Methods are described fine enough.

For more details see the general comments to the authors.

Validity of the findings

The findings are totally valid, the data robust and the conclusion relevant. I pointed at a number of minor issues in the general comments to the authors, but I believe none of these issues affect the fundamental findings of this research.

Additional comments

I really enjoyed your manuscript and I am looking forward both to see the next steps of this research (multiplexing, more complex assemblages) and to potentially apply this approach to some of our questions. And of course I look forward to see the final version of this manuscript published.

Good luck for the revisions!

L. 36 rationale (typo)

L. 51-52 I could not find Mercier and Hamel 1996, but I could also not find any other scientific report on how giant clams act as a shelter for a large number of orgnanisms. From my experience, they do indeed host a few organisms and their shell can be colonized by a variety of organisms, but I don’t think they are more of a shelter than any rubble (if we remove giant clams for the reef, will many organisms lose their shelter? Personally I think not and I would recommend to delete this statement, but if yes (I think some of the co-authors are more specialists of giant clams) then I would recommend to add some more accessible reference maybe.

L. 55 I am a bit surprised to see a reference to an article from a German aquarist magazine here. Did you really read this reference and are you sure it is appropriate? (Same comment overall for the first part of the introduction). In this specific case, I would rather recommend referring to one or few of the following: Romance and Grossman 1989 Palaios, Watanabe and Oba 1999 Journal of Geophysical Research: Oceans, Watanabe et al. Palaeogeography 2004 or other relevant papers as there are quite a lot of research done in this field.

L. 56-58 Maybe split this sentence in two? I am not sure how the “as a highly prized resource” relates to “the habitat degradation”. Actually the simple way may be to remove the “as” in the beginning of the sentence. Or split it, and in this case you could define a bit better the value of giant clams (i.e. as food and recently as jewelry in China for example). Actually, my personal feeling is that the economic value of the giant clam is much more important than their ecological role in the reef, maybe this part could be first in the paragraph (again I am no giant clam specialist).

L. 74 It is up to the authors but personally I am not sure Shinzato et al 2018 needs absolutely to be referred to. Nowadays any research can be published without meaning it is good research and Shinzato et al 2018 is a perfect example of combined poor methodology and overstatements (the title with “zoothantellae”already illustrates the attention given by authors, reviewers and editors to the manuscript, but aside from this typo there are I think enough approximations in this paper to encourage me to classify it in the category “papers to forget”). That being said, this is my personal opinion and I let you free to decide.

l.80 to 105: Personally I would be very interested to know why you selected this approach rather than the the Stat et al. 2017 method, is it impossible/difficult to adapt the fusion primer approach to multiple markers. I can imagine troubles at the creation of the first demultiplexing on the illumina machine, but if you thought about it already, it might be nice to explain it a bit so readers like me will save some time thinking how to use fusion approach with multiple markers. At the moment I did not find a solution, so I can easily imagine you have good reasons to use the Nextera approach…

L. 111 Archipelago? (Without “s”)

L. 164 this is again more a personal style comment, but I would remove the “unprecedented” in this sentence as it sounds a bit like you are a “carpet merchant” try to sell some products. “… provides speed and accuracy (…) without the need for time-consuming alignment steps.” Is a good enough reason to use this software and sounds more objective.

L. 181 Please indicate the BLASTn parameters (at least e-value threshold) as it is important to assess the reliability of the BLAST assignment.

L. 208 According to your text, this is the raw read output. To me this seems a very low output for a 500-cycle MiSeq Reagent Kit v2.

L. 227 I am not sure to understand why your Symbiodiniaceae reference was so incomplete? Why were those 27 genotypes no in your database? Are they unreliable? I guess there is a good reason as it would be very easy to append your reference database, but I cannot understand it clearly based on the text. Maybe it is related to the next sentence, but please clarify (how many of the 11 retained sequences were in your database?). Similarly I am confused why, based on Fig.2 it seems your reference data base did not include C1 as this one appears to have been identified through NCBI…

L. 248 This is the only “red flag” I see in the manuscript. Both plants in the example (two North American/Canadian plants have nothing to do with Polynesian coral reefs and are obvious contaminations, and possibly misidentifications. While this does not absolutely affect the general findings of the manuscript, these findings indicates that the limit of reliability of the approach has been reached and probably this percentage of reads is just trash. And this should be considered and discussed. Even for an eDNA study I would be very suspicious to find DNA of these plants in Polynesian waters, so the fact that it is present in Tridacna biopsies... Please, although it seems an increasing trend, don’t produce one of these metabarcoding paper where authors don’t know what are the organisms sequenced and blindly trust bioinformatics, especially as your manuscript is a very nice little study making its point quite well I think.

L. 269 related to the previous comment, please don’t tell me you are suggesting that terrestrial north American plants are host-associated organisms. I think it is just a misformulation or a misunderstanding from my part as the generation of scientists that cannot use its brain to interpret bioinformatic results is coming rapidly enough without our generation needing to show the example.

L. 277-278, I would remove the first part of the sentence as the money values are constantly changing and very country dependent (with the same methods, it is certainly not the same price per sample in Australia, USA, Japan or Indonesia). I would just state “The costs depend on the number…”

L. 283-288 While you are at the future perspectives, I would also suggest to list that this approach succeeded its first test in a low diversity setting (mainly A., with a bit of C. and a tiny bit of G.), the next step would be to test if higher diversity and more equal proportions in the samples will introduce more bias.

Reviewer 3 ·

Basic reporting

Overall I thought this was a high quality paper. The introduction provided a clear justification for the research and I was able to follow all sections with ease. In particular, I found their descriptions of dual-index sequencing, and the novel contribution their study makes, very lucid. The standard of writing was high, with very few errors (see General Comments). I was able to access all raw sequence files and it was simple to relate them back to the sample sites described in the paper.

However, I do think Figure 2 could be improved. I had to look at this figure for a long time before I understood all the information it was providing. I think the main problem is that the lines separating different sub-genera on the outer circle are very faint, which makes the clade designation labels and accession number arrows a lot more confusing than they need to be. For example, in the ITS2 circle, the line separating A3 and A6 on the outer ring is nearly impossible to see, and I cannot tell at all where C1 ends and the NCBI cluster (HG942434-AB294624) begins. Could these lines be made black or clearer in some other fashion in order to make the proportions of different sequences recovered more understandable?

Also, as an aside, I’m curious as to why the nearly all the Cladocopium sequences for the ITS2 region, and every Cladocopium sequence for the LSU, was identified with the NCBI database and not the in-house reference databases. Is this because the in-house databases were not particularly comprehensive? I think a comment from the authors to explain this would be useful here.

In addition, this is not necessary but I wonder if the authors would consider making Table S4 a primary table in the manuscript. I personally found it very compelling evidence that the pooled samples recovered very similar sequence proportions to the controls and went a long way to convincing me of the efficacy of this approach.

Experimental design

The work carried out was clearly justified, and suitable for beginning to answer the question they posed. However, given the novelty of this approach, I would note the following points:

I would strongly encourage/expect the authors to provide the full code they used in their bioinformatic pathway, as this would improve reproducibility and encourage other authors to utilise this approach.

An additional supplementary file regarding the design of the new primers (Line 119) would be useful, particularly how they were designed to avoid cladal bias, given the highly variable proportions of different genera recovered using the different markers for S147, S150 and S152 in Figure S2.

Validity of the findings

In general, their conclusions are justified based on the data they produced, and I am satisfied that they have provided proof-of-principle for this technique. However, I have two concerns that should be addressed further:

I think it is important for the authors to add a caveat regarding the decrease in sequencing depth with their approach. It is clear there are generally far fewer sequences recovered per pooled gene than in the control samples (Table 1), something that is not mentioned at all. While this doesn’t matter for their own study, it may have more of an impact for future authors who want to have a complete overview of fine-scale diversity, or study potential low-frequency intragenomic variants etc.

Lines 275-288: It may be a slightly sensitive subject, but given the emphasis that has been placed on “considerable cost saving”, beginning in the abstract and throughout the manuscript, I expected a more detailed and tangible demonstration of this (e.g. “using traditional dual-index HTS methods the sequencing we achieved would have cost $XX, but with the methods utilised in this study it cost $YY.”) Even if the authors do not want to mention specific costs, some relative measure would be useful for readers to assess the strongly promoted significant cost savings, and weigh it against potential negatives such as decreased sequencing depth (e.g. was this method twice as cheap? 1.4 times cheaper?). This seems to be one of the major benefits of this technique that the authors want to push, so I think it needs to be fleshed out slightly to justify the emphasis placed on it.

Additional comments

I have a few small comments that I believe would enhance the quality of the manuscript.

Line 36: ‘rational’ should be ‘rationale’

Lines 56-58: I know what the authors mean, but this sentence is a bit ambiguous, as it could imply that there are endosymbionts inside giant clams that are endangered (rather than the clams themselves). Could be improved with e.g. “… range, the family Tridacnidae also contains…”

Lines 84-91: This is a useful and concise summary – excellent.

Line 130: How was the size of amplicons assessed?

Line 164: I’m not sure about the use of the word ‘unprecedented’ as the only description of Kallisto. I don’t disagree with the authors, I just think an extra line on how it is unprecedented would aid readers who may be less familiar with the latest bioinformatic techniques/literature.

Lines 173-175: I am not very familiar with this approach, so it may be just a lack of knowledge on my behalf, but this sentence is not clear to me. Is it possible to expand slightly to better explain this analytical step?

Line 180: spelling mistake in Symbiodiniaceae

Lines 263-275: Could this discussion be added to regarding the results of Figure S2, which show markedly different proportions of symbiont genera recovered between the 3 gene regions for samples S147, S150 and S152? What are the implications for interpretation when this is the case?

Line 307: missing full stop

Line 310: missing italicization

Line 320-322: This sentence could be rewritten, particularly regarding the awkward use of ‘noteworthy’ at the start, and the double emphasis on those genotypes having not been previously reported (only one of “yet” and “before” is needed).

In the first sheet of Table S3, it is not clear what the column ‘EE’ is – can this be clarified?

It may be just the program I used to open it, but Figure S1 appeared to be cut off and I could only view a small portion of it (Figure S2 was fine though)

Regarding Table S1, I don’t think the presentation of date is self-evident; could signpost better in caption: “…date collected (Year-Month), …”

Regarding Table S2, a very tiny thing, but the ITSD_illu primer name is missing the underscore.

---

## Round 0.2 · accepted · Accept

I am satisfied with all of the corrections made to the manuscript. With regards to the code, since it is not a bioinformatics tool (i.e. the workflow is not the product) PeerJ does not require the code as a criterion for publication. If you would like to add a link later in the interests of further openness, you can add it in the links section of the article page.

# ·

Basic reporting

no comment

Experimental design

no comment

Validity of the findings

no comment

Additional comments

Thank you for addressing all my comments and concerns. I think the manuscript is well improved and ready to go!

Cheers for the good work!

Reviewer 3 ·

Basic reporting

I am impressed with the efforts the authors have taken with their revisions of the manuscript; all my (reasonably minor) concerns have been addressed. This was already a high-quality paper and it is now especially so. It is unfortunate that the bioinformatic pathway could not be provided at this time, but I am happy with the suggestion the authors make regarding it in their rebuttal document. I am pleased to be able to recommend this paper for publication.

Experimental design

No additional comments

Validity of the findings

No additional comments

Additional comments

No additional comments